# The Use of Different Modes of Post-Activation Potentiation (PAP) for Enhancing Speed of the Slide-Step in Basketball Players

**DOI:** 10.3390/ijerph17145057

**Published:** 2020-07-14

**Authors:** Mariola Gepfert, Artur Golas, Tomasz Zajac, Michal Krzysztofik

**Affiliations:** 1Institute of Sport Sciences, The Jerzy Kukuczka Academy of Physical Education in Katowice, Mikolowska 72a, 40-065 Katowice, Poland; m.gepfert@awf.katowice.pl (M.G.); a.golas@awf.katowice.pl (A.G.); 2Human Performance Laboratory, The Jerzy Kukuczka Academy of Physical Education in Katowice, Mikolowska 72a, 40-065 Katowice, Poland; t.zajac@awf.katowice.pl

**Keywords:** performance, conditioning activity, assisted, resisted

## Abstract

*Background*: This study aimed to determine whether 5% of body mass-resisted or assisted conditioning activity (CA) can enhance 5 m slide-step movement performance. *Methods*: Sixteen division I basketball players participated in this study (23.6 ± 4.4 years; 86.3 ± 5.9 kg; 192.3 ± 6.2 cm; training experience 6.7 ± 2.6 years). The experiment was performed following a randomized crossover design, where each participant performed two different exercise protocols—assisted and resisted CA each consisting of four sets of 10 m slide-step movement with 5% of body mass external load and 1 min rest intervals between. To assess the differences between baseline and post-assisted, as well as post-resisted CA, the participants performed a 2 × 5 m slide-step movement 6 min after each CA protocol. The differences in time between baseline, post-assisted and post-resisted CA were examined using repeated-measures ANOVA. *Results*: ANOVA indicated a statistically significant difference between baseline and post-assisted postactivation performance enhancement (PAPE) (*p* = 0.011). There were no significant intragroup differences between baseline and post-resisted CA (*p* = 0.230). *Conclusion*: Findings of the study show that a light load assisted CA (5% of body mass) effectively elicits a potentiation response among basketball players.

## 1. Introduction

Basketball is one of the fastest team sports played worldwide both competitively and recreationally [1,2,3,4,5]. Basketball is characterized by actions involving sprints, dunks and blocked shots [6], lateral movements, jumping and landing, interspersed with frequent and sudden changes of direction, decelerations, and stops [7,8,9]. This means that basketball players need great athletic ability in order to most proficiently demonstrate the speed, strength and power output required to produce a successful basketball performance [10]. Many of the key actions performed by basketball players in a game are based on horizontal movements (sprints and changes of direction), vertical movements (jump shots and rebounds) and combinations of movements on both of these planes, mainly when penetrating to the basket and blocking a shot [11]. These high-intensity movements are usually performed intermittently throughout the game [6]. At the elite level, in an official game, players usually perform numerous short accelerations and decelerations, close to 1000 changes of pace and movement direction. It has also been determined that 30% of game time the players move sideways, utilizing the defensive slide-step [12]. The defensive stance is used to prevent offensive players from moving the ball forward and scoring. Moreover, the defender must also react when the opponent starts offensive action by changing his position using slide-step. Change of direction is associated with such factors as to decelerate, reverse, or change the direction of movement and accelerate [13]. Therefore, the slide-step is one of the most important forms of movement in basketball, especially during defensive actions [14]. Single slide-step actions last between 1–4 s, yet are repeated every 20–30 s, and appear more frequently in close games with an aggressive defense.

As has been mentioned above, the characteristics of basketball require from players’ high levels of neuromuscular abilities (i.e., power output, strength, speed) [15]. Thereby, strength and power training programs with basketball players should focus on developing the performance of explosive movements [16]. An acute enhancement in explosive sport tasks (e.g., throws, sprints, jumps) can be achieved due to the muscular phenomenon called postactivation potentiation (PAP) and recently, an alternative term has been proposed, a postactivation performance enhancement (PAPE). Both lead to sports performance increment; however, differences exist in underlying physiological mechanisms and in the time of enhancement persistence. The primary mechanism underlying PAP is the phosphorylation of the myosin regulatory light chain and a very short period of time (<5 min), while PAPE would be associated with other potential mechanisms (e.g., muscle temperature, water content) and a longer “window of opportunity” (>5 min). In training practice, the potentiation can be achieved by the execution of biomechanically similar conditioning activity (CA) at maximal or near-maximal intensities prior to the subsequent athletic task (e.g., heavy-load back squat before countermovement jump) [17]. Over the past years, there has been a large amount of scientific research related to the use of various CA exercises to enhance the performance of movements such as lifting of free weights [18,19,20], performing ballistic and plyometric exercises [21,22], and maximal isometric voluntary contractions [23]. Although that assisted [24,25] and resisted sprint modalities [26,27,28] are the two commonly used filed methods of improving acceleration and velocity of sprint, the available data regarding their use as CA to induce potentiation of subsequent performance is scarce. Both methods provide specific adaptations—an assisted sprint involves an overspeed stimulus while resisted sprint provides an overloaded stimulus to an athlete. The resisted sprint training includes gravity-resisted modalities, such as sled pulls, elastic cord resistance, parachute or weighted vest running and the assisted sprint training consists of downhill and elastic cord assisted running or assisted towing [26,28,29]. It has been shown that training programs addressing the long-term application of these methods lead to significant improvements in the sport-specific performance variables of acceleration and velocity [30,31,32,33]. However, to the best of our knowledge, only few studies assessed the acute effects of resisted or assisted sprints on subsequent athletic performance [34,35,36,37,38], and just one directly compare effectiveness of resisted versus assisted sprints [39]. Whelan et al. [34] reported a lack of post-resisted sprints with regard to sprint potentiation in active males. Although, a study by Winwood et al. [35] and Seitz et al. [36] observed a significantly faster, 15 and 20 m, respectively, sprint performance enhancement after completing heavy sled pulls with a load of 75% body mass in resistance-trained rugby athletes. Moreover, a study by Smith et al. [37] examined the effectiveness of sled pulls at either 0%, 10%, 20%, and 30% of participants’ body mass on 40 yard sprint performance and found that heavier resistance leads to greater improvements. Further, Tillaar and Heimburg [39] found that only resisted sprints (absolute load of 5 kg for all participants) resulted in faster running time in the 20-m sprint in experienced female handball players. Conversely, assisted sprints (absolute load of 40 kg for all participants) did not cause any performance time changes to normal sprinting. Additionally, Nealer et al. [38] showed that 30% of body mass-assisted CA leads to a significant decrease in sprint time for the first 5 m among female soccer players.

Although research has shown that the various CA modalities significantly increase the performance of different sprint lengths, no data were accumulated to analyze the potentiation effect of more functional activities such as the slide-step movement. Further, according to the authors’ knowledge, the presented study is the second one which directly compares the effectiveness of resisted and assisted CA, and the first one which analyzes the potentiation effect on slide-step movement among basketball players. Thus, the purpose of the present study was to determine whether 5% of body mass-resisted or assisted four sets of 10 m slide-steps can enhance 5 m slide-step movement performance. Based on previous findings, we hypothesized that the resisted slide-step, as well as assisted slide-step CA, will improve the 5 m slide-step movement performance.

## 2. Materials and Methods

### 2.1. Participants

Sixteen division I basketball players participated in the study (age = 23.6 ± 4.4 years, body mass = 86.3 ± 5.9 kg, body height = 192.3 ± 6.2 cm, training experience = 6.7 ± 2.6 years). All players participating in the experiment had valid medical examinations and showed no contraindications to participate in physical fitness tests. The athletes were instructed to maintain their normal dietary habits over the course of the study and not to use any supplements or stimulants for the duration of the experiment. Furthermore, they were informed verbally and in writing about the experimental protocol, the possible risks and benefits of the study, and the possibility to withdraw at any stage of the experiment. All participants gave their written consent for participation. The study protocol was approved by the Bioethics Committee for Scientific Research (10/2018), at the Academy of Physical Education in Katowice, Poland, and performed according to the ethical standards of the Declaration of Helsinki, 2013.

### 2.2. Procedures

In a randomized, crossover fashion, the participants performed two experimental sessions in the 2 × 5 m slide-step test following two different CA modes. Both experimental sessions were carried out always at the same time of a day (between 5 and 7 p.m.) to avoid any effect of circadian rhythm and because training sessions typically took place at this time. The experimental sessions were separated by a week (on Mondays), after 2 days of rest to avoid the effects of fatigue on the results of speed tests. The tests were preceded by a 15 min warm-up, which included 5 min of jogging, dynamic stretching, and several explosive activities (accelerations, changes of movement direction, and slide-stepping). During one experimental session, the players performed the slide-step trial twice, at baseline, after the warm-up and following CA, which included 4 × 10 m of the slide-stepping with horizontally applied assisted external load equaling 5% of body mass, which allowed for a supramaximal speed of movement. This value of external load was chosen since a prior study found that similar resistance (4.7% of body mass) increases the kinematics variables of sprinting [30]. To standardize the stimuli within groups, 5% of body mass was chosen for resisted CA. The participants were instructed to alternate the lead leg in successive repetitions. The load was applied individually using the 1080 Sprint device (1080 Motion, Stockholm, Sweden). This apparatus has been previously used to provide resistance during running in recent studies among athletes [40]. Each player performed 4 sets of 10 m of the slide-stepping, interspersed with 1 min rest intervals in between. A 6 min rest interval was used for each CA before the speed slide-step was repeated. This duration of the rest interval was applied since previous studies showed that the PAP effect peaks at around 4–8 min after the CA [41] and its effectiveness in inducing potentiation of running performance was proven in prior studies [39]. The slide-step test was performed on a wooden basketball floor, and required one change of movement direction, preceded by a touch of a cone by the outside hand. The test was started to form a basketball defensive stance, with the lead foot placed 30 cm before the starting line. The running times were measured by two pairs of dual-beam photocells Witty Gate (Microgate, Bolzano, Italy) timing gates with the measuring precision of 0.01 s. This technology has been previously used to record the time of short sprint results in scientific research [42,43]. On the second occasion, the procedure was identical, except for the mode of CA, which consisted of 4 sets of 10 m of the slide-stepping with resisted external load equaling 5% of body mass.

### 2.3. Statistical Analysis

The Shapiro–Wilk, Levene and Mauchly’s tests were used to verify the normality, homogeneity and sphericity of the sample’s data variances, respectively. Verifications of the intra- and interdifferences between baseline and postpotentiation (assisted and resisted CA) time measures in PAP-Resisted and Assisted were verified using ANOVA with repeated measures. Effect sizes were reported where appropriate and 95% confidence intervals were also calculated. According to Hopkins guidelines, the effect size (eta-squared; η2) was established as follows—0.01, small; 0.06, medium; and 0.14, large. Statistical significance was set at *p* < 0.05. All statistical analyses were performed using Statistica 9.1 (TIBCO Software Inc., Palo Alto, CA, USA) and Microsoft Office (Redmont, Washington, DC, USA), and are presented as means with standard deviations.

## 3. Results

The intra-measures ANOVA analysis revealed significant differences between baseline and assisted CA with *p* = 0.011 and with a large effect size η2 = 0.19 (Table 1, Figure 1). There were no significant differences between baseline and postintervention after the resisted CA (*p* = 0.230), and the effect size was small η2 = 0.04.

The same analysis for inter-measures revealed significant differences between assisted and resisted CA in the slide-step with *p* = 0.031 and with a medium effect size η2 = 0.071. There were no significant intragroup differences between assisted and resisted CA at baseline (*p* = 0.859) and the effect size was very small η2 = 0.003.

## 4. Discussion

The main finding of the study was that only assisted CA significantly enhanced the speed of the slide-step movement, among basketball players. This study revealed that a 5% body mass-assisted slide-step significantly decreased the time during the following two sets of 5 m slide-step movement by ~4.6% (from 3.24 ± 0.15 to 3.09 ± 0.16 s). On the contrary, a 5% body mass-resisted CA did not cause any changes in slide-step performance compared to baseline. Thus, the results of the presented study indicate that the assisted CA effectively potentiates performance of the basketball sport-specific task. Moreover, the results indicate that 5% body mass-assisted activation is sufficient to acutely improve the performance of the slide-step movement among basketball players. 

To the best of our knowledge, there is no available data regarding acute speed changes of the slide-step movement after resisted and assisted CA, which limits the possibility of comparing our results with other studies. Nevertheless, several studies examined the influence of resisted CA, and one actually compares these types of CA on sprint performance. Therefore, significant knowledge and training guidelines can be derived from the current study. The presented results are inconsistent with previous studies that examined the effectiveness of resisted CA to induce performance potentiation. A study by Seitz et al. [36] reported a significantly faster 20 m sprint time following sled pulls with a load of 75% body mass in resistance-trained rugby players. Winwood et al. [35] showed a significant improvement of 15 m sprint time after a similar stimulus. However, Tillaar and Heimburg [39] found that resisted sprints with an absolute load of 5 kg, which corresponded to approximately ~7.3% of average subjects’ body mass, resulted in faster running time in the 20 m sprint among experienced female handball players. Wilson et al. [44] reported that several factors such as training experience and strength level of athletes, as well as the type of CA, volume, intensity, and the rest interval between the CA and the explosive movement, modulate the magnitude of potentiation. The explanation for contradictory results between our study and previous investigations may be related to the implemented intensity of the CA. Taking into consideration the strength level of the athlete, a higher intensity of the CA may be needed to elicit the potentiation effect [45]. Further, Smith et al. [37] indicated that 30% of body mass-resisted CA lead to greater improvement of the 40-yard sprint when compared to 10% loading. According to the findings revealed by Smith et al. [37] that heavier resistance sled pulls lead to greater speed improvements, and considering that the participants of this study had extensive training experience, it can be concluded that 5% of body mass-resisted CA was insufficient to induce potentiation of subsequent performance in the slide-step movement of basketball players. Based on the results of previous studies, the use of resisted CA may require the implementation of heavy-loads (> 30% of body mass) among male athletes. 

A novel finding of this study was that 5% of body mass-assisted CA significantly enhanced the performance of the slide-step movement. According to the best of authors’ knowledge, the presented study is only the second one which directly compared the effectiveness of resisted and assisted CA in eliciting potentiation. To date, a majority of scientific research was focused on assessing the effectiveness of resistance exercises in eliciting potentiation of following sports-tasks, while the interest of use more practical modalities such as plyometric exercises and “over-speed” conditions has risen recently [21,39,45]. A lack of studies that analyzed the efficiency of assisted CA may be related to the fact that athletes have to “handle” the higher than voluntarily achieved velocities of running, which means that an experienced and powerful group of subjects is required. A study by Tillaar and Heimburg [39] revealed that the assisted CA did not cause any performance time changes in comparison to normal sprinting, while resisted CA enhanced running time in the 20 m sprint. On the contrary, our results showed that 5% of body mass-assisted CA was superior in eliciting potentiation in the 5 m slide-step movement in comparison to resisted CA. This discrepancy may be associated with a lack of experience with assisted sprinting among subjects that participated in the study of Tillaar and Heimburg [39]. In the current study, athletes were familiarized with the use of assisted conditioning exercises, while for the participants of the Tillaar and Heimburg [39], investigation in the task was completely novel. Moreover, the authors used an absolute load of 40 kg which corresponded to approximately ~58% of average subjects’ body mass. Clark et al. [30] showed that a towing resistance greater than 3.8% of body mass had a negative effect on the running mechanics of the athlete. Since the participants of the Tillaar and Heimburg study [39] were not familiarized with this type of movement, this may also explain the contradictory findings. Additionally, it can be suggested that the rationale for potentiation may be attributed to increased neural activation of higher-order motor units, as was found by Mero and Komi [45] after supramaximal sprints. 

The present study has some limitations which have to be addressed. Although the results showed that the potentiation occurred during the slide-step movement after assisted CA, the direct causes of these changes cannot be determined and explained due to the lack of physiological analysis. Furthermore, neither the electromyography of the involved muscles nor the kinematics were investigated. Moreover, the assessment of performance changes was made only on the basis of a single value of resistance (5% of body mass). Therefore, future studies are required to assess the acute impact of different resisted and assisted loads used as CA with different volumes and rest intervals on the potentiation effect.

## 5. Practical Implications

The assisted CA with 5% of body mass resistance is sufficient to significantly enhance the slide-step movement performance among basketball players. However, the use of resisted CA with a load of 5% body mass may be insufficient to induce potentiation. A proposed training strategy might be of interest to coaches and practitioners. Due to the lack of equipment requirements (e.g., barbell, free weights), the employment of assisted CA as part of a warm-up routine may be an attractive and easy solution to induce acute performance enhancement before the competition. Moreover, a repeated acute improvement of performance following assisted CA may represent a stimulus for athletes attempting to enhance slide-step performance. However, to assess the validity of that training modality, there is a need to conduct long-term studies that evaluate the effectiveness of it to enhancing running performance.

## 6. Conclusions

Basketball, as a complex sport, requires a high level of neuromuscular fitness, especially power, strength, as well as speed and agility [15]. Findings of the current study show that a light resistance-assisted CA (5% of body mass) effectively elicits a potentiation response among experienced basketball players. Nevertheless, it can be speculated that the resisted CA requires a significantly higher external resistance to induce potentiation of subsequent sport-specific tasks, such as the basketball slide-step.

## Figures and Tables

**Figure 1 ijerph-17-05057-f001:**
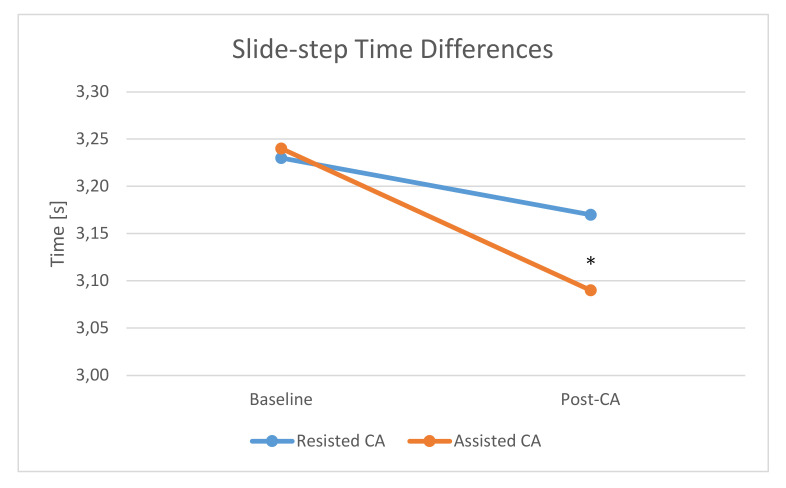
Differences in slide-step performance between baseline and postactivation protocols. * Statistically significant differences *p* < 0.05.

**Table 1 ijerph-17-05057-t001:** Slide-step results at baseline and postactivation protocols.

Variable	Baseline	RESISTED CA	ES	Baseline	ASSISTED CA	ES
Timing (s)	3.23 ± 0.15 (3.15 to 3.31)	3.17 ± 0.13 (3.09 to 3.24)	0.04	3.24 ± 0.15 (3.16 to 3.32)	3.09 ± 0.16 * (3.01 to 3.18)	0.19

Mean ± standard deviation (SD); * statistically significant differences *p* < 0.05; CA—conditioning activity; ES—effect size η2.

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
