# Peer review of "The Use of Different Modes of Post-Activation Potentiation (PAP) for Enhancing Speed of the Slide-Step in Basketball Players"

_ijerph, 2020, doi:10.3390/ijerph17145057_

Round 1
Reviewer 1 Report
#Comments to authors
The manuscript, entitled “The use of different modes of post-activation potentiation (PAP) for enhancing the speed of the slide-step in basketball players,” aimed to determine whether 5% of body mass resisted or assisted 4 sets of 10m slide-steps can enhance 5m slide-step movement performance. The study hypothesized that resisted slide-step, as well as assisted slide-step CA, would improve 5m slide-step movement performance. The study has some interesting academic merits. However, the authors should address the following comments in order to increase readability among international readers whose major are not in sports or basketball.
- Procedures:
- In the experiment, the authors required the participants to perform 2 experimental sessions at 5 and 7 p.m. It would be better if the authors explained why this time was best appropriate for the experiments.
- Likely, why 5% f the body mass was the option?
- Clarification of the procedures could make your experimental processes easier to understand.
- Furthermore, the time interval for the rest was 6 min. What were the reasons to decide 6 min for the rest?
- Table1 is messy.
- Discussion
- The authors explained that a 5% body mass assisted slide-step significantly decreased the time during 2 sets of 5 m slide-step movement. The readers would love to see a specific estimated time that it could decrease.
- It would be more interesting if the authors explained the implications of your findings. For example, the study claimed that 5% of body mass assisted CA significantly enhanced the performance of the slide-step movement. However, the use of resisted CA with a load of 5% body mass may be insufficient to induce potentiation. What is your suggestion to enhance the basketball players’ performance based on your experimental procedures and findings?
- Line 193-195: The authors claimed that this study is one of the only two available papers which directly compared the effectiveness of resisted and assisted CA in eliciting potentiation. Therefore, I would love to hear some reasons why previous studies did not use 5% of body mass resisted and assisted CA to test the basketball players’ performance and what would be the reasons why previous studies did not interest in comparing the effectiveness of resisted or assisted CA. Such an explanation may add some extra merits to your findings.
Author Response
We would like to thank the Reviewer for the time devoted to evaluating the manuscript and for providing critical comments as well as suggestions that helped us improve our paper. Your remarks are an invaluable lesson for us, and we will take full advantage of it when planning our next research project covering a similar subject matter. We hope that the revised manuscript satisfactorily addresses all the raised issues. The responses to the Reviewer’s comments are below.
Procedures:
In the experiment, the authors required the participants to perform 2 experimental sessions at 5 and 7 p.m. It would be better if the authors explained why this time was best appropriate for the experiments.
Reply: According to Reviewer suggestion, following sentences have been added: Line: 111-115: “Both experimental sessions were carried out always at the same time of a day (between 5 and 7 p.m.) to avoid any effect of circadian rhythm and because training sessions typically took place at this time. The experimental sessions were separated by a week (on Mondays), after 2 days of rest to avoid the effects of fatigue on the results of speed tests.”
Likely, why 5% f the body mass was the option?
Reply: The rationale for this choice has been provided: Line: 120-122: “This value of external load was chosen since a prior study found that similar resistance (4.7% of body mass) increases the kinematics variables of sprinting [30]. To standardize the stimuli within groups, also 5% of body mass was chosen for resisted CA.”
Clarification of the procedures could make your experimental processes easier to understand.
Furthermore, the time interval for the rest was 6 min. What were the reasons to decide 6 min for the rest?
Reply: As suggested, the rationale for this choice has been provided: Line 127-130: “This duration of rest interval was applied since previous studies showed that PAP effect peaks at around 4-8 min after the CA [41] and its effectiveness in inducing potentiation of running performance was proven in prior studies [39].”
Table1 is messy.
Reply: Table 1 has been fixed.
Discussion
The authors explained that a 5% body mass assisted slide-step significantly decreased the time during 2 sets of 5 m slide-step movement. The readers would love to see a specific estimated time that it could decrease.
Reply: According to Reviewer suggestion, following sentences have been added: Line: 166-168: „This study revealed that a 5% body mass assisted slide-step significantly decreased the time during the following 2 sets of 5 m slide-step movement by ~4.6% (from 3.24 ± 0.15 to 3.09 ± 0.16s).”
It would be more interesting if the authors explained the implications of your findings. For example, the study claimed that 5% of body mass assisted CA significantly enhanced the performance of the slide-step movement. However, the use of resisted CA with a load of 5% body mass may be insufficient to induce potentiation. What is your suggestion to enhance the basketball players’ performance based on your experimental procedures and findings?
Reply: As suggested, the practical implications session has been amended: Line: 232-240: “However, the use of resisted CA with a load of 5% body mass may be insufficient to induce potentiation. A proposed training strategy might be of interest to coaches and practitioners. Since, due to the lack of equipment requirements (e.g. barbell, free weights), the employment of assisted CA as part of a warm-up routine may be an attractive and easy solution to induce acute performance enhancement before the competition. Moreover, a repeated acute improvement of performance following assisted CA may represent a stimulus for athletes attempting to enhance slide-step performance. However, to assess the validity of that training modality, there is a need to conduct long-term studies that evaluate the effectiveness of it to enhancing running performance.”
Line 193-195: The authors claimed that this study is one of the only two available papers which directly compared the effectiveness of resisted and assisted CA in eliciting potentiation. Therefore, I would love to hear some reasons why previous studies did not use 5% of body mass resisted and assisted CA to test the basketball players’ performance and what would be the reasons why previous studies did not interest in comparing the effectiveness of resisted or assisted CA. Such an explanation may add some extra merits to your findings.
Reply: According to Reviewer suggestion, a possible cause of this issue has been provided: Line 201-206: “To date, a majority of scientific researches was focused on assessing the effectiveness of resistance exercises in eliciting potentiation of following sports-tasks, while the interest of use more practical modalities such as plyometric exercises and “over-speed” conditions has risen recently [21,39,45]. A lack of studies that analyzed the efficiency of assisted CA may be related to the fact that athletes have to “handle” the higher than voluntarily achieved velocities of running, which means that experienced and powerful group of subjects are required.
Reviewer 2 Report
I enjoy reading the paper and I have not technical comments. In my opinion, the paper is suitable for publication in IJERPH MDPI.
Just a minor comment to improve the quality of the manuscript. This paper is roughly 6 pages long (with no reference). Usually, IJERPH papers are longer so that I would suggest to add some figures in order to increase the reader's interest in the paper.
Change it to 'minor revision'after I add the following further comments
Please clarify the criteria behind all the choices made by the authors in their experiments. All of them seem to be arbitrary. Please specifically address the reason behind any choice. It gives more value to the paper. For instance, on the basis of what the time of experimental session is chosen? why the paper refer to 5% of the mass body? (why 5 and not, 3, 7 or 8 ?) Why 6 min or rest (and not 5 or 10) ?
Author Response
We would like to thank the Reviewer for appreciating our work and for their positive comments. We are also grateful for the Reviewer taking the time to evaluate this manuscript and for providing comments as well as suggestions that helped us improve the quality of our paper. We hope that the revised manuscript satisfactorily addresses all the raised issues. The responses to the Reviewer’s comments are below.
I enjoy reading the paper and I have not technical comments. In my opinion, the paper is suitable for publication in IJERPH MDPI.
Just a minor comment to improve the quality of the manuscript. This paper is roughly 6 pages long (with no reference). Usually, IJERPH papers are longer so that I would suggest to add some figures in order to increase the reader's interest in the paper.
Change it to 'minor revision'after I add the following further comments
Please clarify the criteria behind all the choices made by the authors in their experiments. All of them seem to be arbitrary. Please specifically address the reason behind any choice. It gives more value to the paper. For instance, on the basis of what the time of experimental session is chosen? why the paper refer to 5% of the mass body? (why 5 and not, 3, 7 or 8 ?) Why 6 min or rest (and not 5 or 10) ?
Reply: According to the Reviewer suggestions, the rationale for the selected variables has been provided through the paper, what significantly extended the length of the manuscript: - for the selected time of experimental sessions: Line: 111-115: “Both experimental sessions were carried out always at the same time of a day (between 5 and 7 p.m.) to avoid any effect of circadian rhythm and because training sessions typically took place at this time. The experimental sessions were separated by a week (on Mondays), after 2 days of rest to avoid the effects of fatigue on the results of speed tests.” The rationale for selected load: Line: 120-122: “This value of external load was chosen since a prior study found that similar resistance (4.7% of body mass) increases the kinematics variables of sprinting [30]. To standardize the stimuli within groups, also 5% of body mass was chosen for resisted CA.” In the case of selected rest interval: Line 127-130: “This duration of rest interval was applied since previous studies showed that PAP effect peaks at around 4-8 min after the CA [41] and its effectiveness in inducing potentiation of running performance was proven in prior studies [39].”
In addition, we would like to mention that the practical implication session has been amended according to Reviewer #1 suggestions: Line 232-240: “However, the use of resisted CA with a load of 5% body mass may be insufficient to induce potentiation. A proposed training strategy might be of interest to coaches and practitioners. Since, due to the lack of equipment requirements (e.g. barbell, free weights), the employment of assisted CA as part of a warm-up routine may be an attractive and easy solution to induce acute performance enhancement before the competition. Moreover, a repeated acute improvement of performance following assisted CA may represent a stimulus for athletes attempting to enhance slide-step performance. However, to assess the validity of that training modality, there is a need to conduct long-term studies that evaluate the effectiveness of it to enhancing running performance.”
Round 2
Reviewer 1 Report
Dear authors,
I could see a significant effort in revising the manuscript. The current form is acceptable for publication.
Best wishes